# Self-Seeding Microwells to Isolate and Assess the Viability of Single Circulating Tumor Cells

**DOI:** 10.3390/ijms20030477

**Published:** 2019-01-23

**Authors:** Kiki C. Andree, Fikri Abali, Lisa Oomens, Fiona R. Passanha, Joska J. Broekmaat, Jaco Kraan, Pauline A.J. Mendelaar, Stefan Sleijfer, Leon W.M.M. Terstappen

**Affiliations:** 1Medical Cell BioPhysics, University of Twente, Hallenweg 23, 7522 NH Enschede, The Netherlands; k.c.andree@utwente.nl (K.C.A.); f.abali@utwente.nl (F.A.); f.passanha@maastrichtuniversity.nl (F.R.P.); 2VyCAP B.V., Abraham Rademakerstraat 41, 7425 PG Deventer, The Netherlands; Lisa.oomens@vycap.com (L.O.); joska.broekmaat@vycap.com (J.J.B.); 3Department of Medical Oncology, Erasmus MC Cancer Institute and Cancer Genomics Netherlands, Dr. Molewaterplein 40, 3015 GD Rotterdam, the Netherlands; j.kraan@erasmusmc.nl (J.K.); p.mendelaar@erasmusmc.nl (P.A.J.M.); s.sleijfer@erasmusmc.nl (S.S.)

**Keywords:** CTC, self-seeding microwells, single cell isolation, breast cancer, prostate cancer

## Abstract

The availability of viable tumor cells could significantly improve the disease management of cancer patients. Here we developed and evaluated a method using self-seeding microwells to obtain single circulating tumor cells (CTC) and assess their potential to expand. Conditions were optimized using cells from the breast cancer cell line MCF-7 and blood from healthy volunteers collected in EDTA blood collection tubes. 43% of the MCF-7 cells (nucleus+, Ethidium homodimer-1-, Calcein AM+, α-EpCAM+, α-CD45-) spiked into 7.5 mL of blood could be recovered with 67% viability and these could be further expanded. The same procedure tested in metastatic breast and prostate cancer patients resulted in a CTC recovery of only 0–5% as compared with CTC counts obtained with the CellSearch^®^ system. Viability of the detected CTC ranged from 0–36%. Cell losses could be mainly contributed to the smaller size and greater flexibility of CTC as compared to cultured cells from cell lines and loss during leukocyte depletion prior to cell seeding. Although CTC losses can be reduced by fixation, to obtain viable CTC no fixatives can be used and pore size in the bottom of microwells will need to be reduced, filtration conditions adapted and pre-enrichment improved to reduce CTC losses.

## 1. Introduction

Assessment of the molecular characteristics of tumor cells from patients is essential for treatment decision making and for research. Currently, tissue from solid tumors for such characterization is obtained through invasive surgical procedures such as taking a biopsy [1,2]. These procedures cannot be performed on a regular basis and for patients with metastatic disease one tumor site may not be representative for all the metastatic sites. Tumor cells continuously change at the molecular level and frequently are associated with the occurrence of resistance to the administrated drugs [3]. Availability of non-invasive methods for the detection and monitoring of cancer is sought after but remains a technical challenge. The presence of tumor cells and tumor nucleic acids in the blood of cancer patients is being investigated for its potential as a non-invasive real time biopsy process. These non-invasive methods can yield information about the genetic profile of cancers and track genomic transformations [1,4,5].

Circulating tumor cells (CTC) are a rare heterogeneous cell population shed by tumors into the blood. CTC can be used to monitor efficacy of therapy and their molecular characterization can improve treatment strategies. For probing the reaction of drugs to CTC one however needs viable CTC. Although the frequency of CTC is known in various cancers [6] little knowledge is available on the proportion of viable CTC [7,8,9]. To date protocols [10] are available for the culture of organoids from tumor biopsies to be used as a model for disease [11,12]. However, culturing CTC has only been demonstrated by a few groups. Kolostova et al. used a size based separation method to isolate and culture urothelial CTC directly on the separation membrane [13]. Second, Cayrefourcq et al. negatively enriched blood samples from 71 patients with metastatic colon cancer patients and successfully established one permanent cell line from a patient having a CTC count of ≥300 [14]. In prostate cancer, Gao et al. succeeded in establishing a 3D organoid system for the long-term culture of CTC derived from peripheral blood of castration resistant metastatic patients [12]. The group of Nagrath focused on lung cancer and developed a novel in situ capture and culture methodology for ex vivo expansion of CTC using a 3D co-culture model. CTC were successfully expended from 14 of 19 early stage lung cancer patients, using a three dimensional co-culture model, including fibroblasts, to support tumor development [15].

In addition, very recently several groups looked into the use of leukapheresis products, as a source for CTC, where the likelihood of finding of CTC is greater and they were able to establish CTC cultures in mouse models [16,17]. These discussed examples indicate that the establishment of functional CTC cell line models is feasible. The isolation and in vitro culture of CTC may provide an opportunity to noninvasively monitor the changing patterns of drug resistance in individual patients while their tumors acquire new mutations and might improve treatment. As methods to expand CTC are still in their infancy. Several key factors have to be considered, no universal recipe for culturing patient derived CTC exists and in fact, each patient’s cell might require slightly different growth conditions. Hence the development and optimization of isolation techniques require a gentle treatment of the cells to incorporate efficient culturing strategies. Also, when interested in CTC heterogeneity what is lacking in the studies discussed above is the possibility to assess molecular heterogeneity between CTC within an individual patient. Cultures are established from the bulk of CTC isolated from patients making it harder to assess their heterogeneity. Single cell isolation techniques might contribute to this demand. Here we use the previously introduced self-sorting microwells [18,19] to establish methods to discriminate between individual viable and non-viable cancer cells and establish conditions to maintain the viability of the cancer cells. We demonstrate a relatively fast (<2 h from whole blood to viable individual CTC) and easy workflow to isolate pure CTC without any background of hematopoietic cells and their subsequent culture. The methods and conditions are optimized using cells from cancer cell lines and tested on blood samples from metastatic breast and prostate cancer patients.

## 2. Results

### 2.1. Cell Viability in Microwells

A cell suspension of MCF-7 cells was stained with Calcein AM (Calc AM) to identify live cells and Ethidium homodimer-1 (EthD1) to identify dead cells and seeded into the microwells. The viability of the cells in the microwells was assessed by fluorescence microscopy. Panel a of Figure 1 shows 30 of the 6400 microwells, 11 wells contained a single viable cell (green dots), one well with two viable cells, one well with one dead cell (red dot) and one well with a viable and dead cell (green and red dot). We observed that the majority of the cells (~95%) was viable directly after seeding into the microwells. Subsequent examination after 4 h, 1 day, 2 days and 4 days, while keeping the microwells in culture conditions, showed a decrease in viability to respectively 88, 65, 57 and 34% (Panel b). As control for the microwell seeding, MCF-7 cells were fluorescent activated cells sorting (FACS) sorted and manually pipetted into a 96 wells plate and viability overtime is illustrated in Figure 1b. Between 0 and 4 h no difference was observed between the three methods, but after 1, 2 and 4 days the pipetted cells showed the least decrease in viability followed by the microwell seeding and FACS sorted cells.

### 2.2. Cell Viability after Punching

The punching efficiency and viability after punching was established by seeding MCF-7 cells into microwells and punching them directly or two days after seeding, into a 96 well culture plate for further expansion. Figure 2 shows typical growth patterns of punched single cells immediately after seeding and cells that were punched two days after seeding. The figure shows the punched cells at day 1, 2 and 8.

The number of cells recovered, the viability and the number of cells that divided were counted. In total more than 80 microwell bottoms containing single viable cells and 78 well bottoms with cells that formed colonies after two days of culture were punched. Microwell bottoms that were detectable in the culture plate after punching were counted as a successful punch. 

The punch efficiency for MCF-7 cells after seeding and immediate punching ranged from 74–90% (mean 82%, SD = 7) and the punch efficiency of MCF-7 cells that divided within the two days after seeding and punched after two days ranged from 68–87% (mean 79%, SD = 7). We observed that the cells that formed colonies were able to migrate inside the wells. This could be an explanation for the lower punching efficiency two days after seeding into the microwells as cells might have migrated to the wall of the microwell and therefore could not be successfully punched.

The effect of punching on the cell viability was assessed by allowing the cells to adhere to the culture plate for 4 h. The number of viable cells, dead cells and cells that divided were counted. For comparison to punching cells, cells were also sorted by FACS and by manually pipetting into a cell culture plate. The highest percentage of living cells were found for FACS (range 89–100%, mean 96%, SD = 4%) followed by punching (range 78–82%, mean 79%, SD = 2%) and pipetting (range 54–94%, mean 78%, SD = 13%). Viable cells were cultured and cell growth was monitored. To assess the growth efficiency after punching, the total number of single cells which formed colonies after 14 days of culture was determined for the three different cell seeding methods. Direct punching of cells resulted in the colony formation ranging from 73–93% (mean 81%, SD = 10%). A decrease in colony formation is found for FACS, range 38–51%, mean 43, SD 4.3 and pipetting, range 52–84% (mean 70%, SD = 11%), Figure 3.

### 2.3. Isolation of Tumor Cells in Whole Blood

The complete workflow of single CTC isolation and identification followed by culture is illustrated in Figure 4. To simulate CTC in cancer patients ~300 pre-stained MCF-7 and ~300 MDA-MB-231 were spiked in two 7.5 mL whole blood aliquots from 10 healthy volunteers. The spiked blood was depleted of CD45+ cells and the suspension was placed on the self-sorting microwells. The MCF-7 and MDA-231 cells present in the microwells were identified by fluorescence microscopy and 53% ± 11 of the MCF-7 cells and 50% ± 11 of the MDA-231 could be recovered. The origin of the cell losses was investigated. Seven experiments were conducted in which pre-stained MCF-7 and MDA-MB- 231 were directly seeded onto the microwells and resulted in a loss of 26% ± 15 for MCF-7 and 18% ± 9 for MDA-MB-231 cells. The loss of MCF-7 and MDA-MB-231 cells by depletion of CD45+ cells through RosetteSep™ was investigated in four experiments in which the pre-stained MCF-7 and MDA-MB-231 cells were measured by flowcytometry in the leukocyte depleted cell suspension and showed a loss of 26% ± 7 for MCF-7 and 29% ± 6 for MDA-MB-231 cells. These results indicated that the ~50% cell loss could be accounted for by both the filtration through the 5 µm pores of the microwells and the leukocyte depletion.

Next, a staining protocol for the identification of the enriched tumor cells was developed and tested. The protocol contained Hoechst to identify the cell nucleus, Calc AM to identify viable cells, EthD1 to identify death cells, CD45-PerCP to identify leukocytes and EpCAM-Alexa647 to identify cells of epithelial origin. To enable the evaluation of the procedure the MCF-7 and MDA-231 cells were prelabelled with CellTrace™ Violet. A typical example of a stained MCF-7 cell in a microwell is shown in Figure 5.

The entire workflow was performed on 7.5 mL blood aliquots from six donors spiked with either ~100 MCF-7 cells or ~100 MDA-MB-231 cells. The efficiency of the method was determined by analyzing the recovery of spiked tumor cells inside the microwells. In addition, the percentage of viable tumor cells and the percentage of cells that showed cell growth after two days of culture was determined. The results are shown in Table 1. In these experiments 85–95% of the 6400 wells are occupied with a cell.

### 2.4. Tumor Cells in Blood of Metastatic Breast and Prostate Cancer Patients

To evaluate whether also CTC from cancer patients could be isolated with this procedure, es established on spiked tumor cells from cell lines, 7.5 mL blood samples collected in EDTA blood draw tubes from five metastatic breast and two prostate cancer patients were processed. Only samples with >10 CTC enumerated using the CellSearch system were used and the results are shown in Table 2. Whereas with MCF-7 cells and MDA-MB-231 cell recoveries of respectively 43 and 48% were obtained in cancer patients, a recovery of CTC of only 0–5% as compared with the number detected by CellSearch was obtained. Possible explanations of this cell loss are a smaller size and greater flexibility of the CTC in cancer patients as compared to the cell lines and a lower expression of the EpCAM antigen on the CTC in cancer patients as compared to the cell lines. To probe whether the recovery of CTC could be improved two blood samples of a prostate cancer patient were processed with the procedure, one with the standard approach and one in which the CD45 depleted sample was permeabilized and fixed to allow for Cytokeratin staining. The CTC recovery improved from 2 to 27%. Whether this improvement is because the cells become more rigid after the fixation and or the higher density of Cytokeratin as compared to EpCAM cannot be concluded from this experiment. 

Enumeration in the microwells and viability was assessed using a microscope. Figure 6 shows an example of three CTC found in a patient blood sample. 

## 3. Discussion

In this study we aimed to develop a workflow where self-sorting microwells are used as a tool to isolate and grow single CTC from blood. To investigate whether cell viability could be maintained within the microwells after seeding and punching we first placed cells from the breast cancer cell line MCF-7 present in cell culture medium on the microwells and showed a single cell sorting efficiency of ~90% with a cell viability immediately after a seeding of 90%. Similar viabilities were obtained after sorting single MCF-7 cells by FACS or by manual pipetting (Figure 1). After 1, 2, and 4 days of culture inside the microwells viability decreased more rapidly in the microwells as compared to manually pipetted cells but less when compared to FACS. Hydrodynamic forces and physical stress endured during the filtration or FACS may account for this difference [18,20,21,22]. For filtration, contact of the cells with the well bottom surface of the microwells made of silicon nitride may also account for the larger decrease in viability as compared to manual pipetting and maybe overcome by applying a cell friendly coating on the well bottom surface before cell seeding. 

Next, we investigated whether the ’punching’ of the cells into wells of microtiter plates effected the viability. A rather good punch efficiency for MCF-7 cells of ~82% was obtained with a viability of ~79%. Viability was comparable with that obtained by FACS (~96%) and pipetting (~78%). The ability of these single cells to form colonies after 14 days of culture was remarkable good in the microwells with ~81% of the MCF-7 cells forming colonies compared to 43% when using FACS and ~70% when using pipetting. The punching method therefore did not affect cell viability and cells were able to expand even when they adhered to the well bottom that came along with the punching (Figure 2). 

To be able to isolate CTC from blood and determine their viability and potential for cell division the number of leukocytes in 7.5 mL of blood is too large to pass through self-sorting microwells containing 6400 individual wells without clogging the pores in the wells. To reduce the number of leukocytes in blood we used the RosetteSep™ leukocyte depletion protocol [7,19]. To fluorescently label CTC. Hoechst was used as a nuclear stain, α-EpCAM-Alexa647 to identify CTC, α-CD45-PerCP to exclude leukocytes and Calc AM and EthD1 to identify the viable and dead CTC. CellTrace Violet labelled cells from the breast cancer cell line MCF-7 and MDA-MB-231 spiked into blood were used to determine the feasibility of the approach. Recovery of ~43% of MCF-7 and ~48% of MDA-MB-231 cells was obtained with a viability of 67 and 70% and cell growth of 59 and 49% of the single cells. Loss of the cells could be contributed equally to the leukocyte depletion and filtration. The slightly larger loss by filtration of the MDA-MB-231 cells as compared to the MCF-7 cells can be contributed to the larger size of the MCF-7 cells (16.3 µm versus 15.6 µm) [22]. 

To evaluate whether this workflow could also be used to identify CTC from patients and determine their viability and capability to grow we processed blood samples from metastatic cancer patients with at least 10 CTC in 7.5 mL of blood by CellSearch. Only ~2% of CTC could be retrieved using this workflow raising the question where the CTC are lost. Possible explanations are the smaller size of CTC as compared to the cells from the cell lines. CTC are generally larger in size then WBC, and this difference is used in the microwell filtration step to remove remaining WBC and RBC after depletion and obtain individual CTC in the microwells. However, where the size of cells from tumor cell lines is known, the actual size of CTC in individual patient is unknown and might cover a wide range [23]. In addition to size, also cell stiffness or rigidity has an important role in passage through a pore [24], cell stiffness of patient CTC may vary and be less than in cell lines, which might explain part of the cell loss. In addition to these physical properties, variation in biological cell properties might also contribute to cell loss. For example, because of lesser staining with EpCAM as compared to Cytokeratin as used in the CellSearch system. In order not to affect viability, EDTA was used as an anticoagulant in contrast to the draw tubes that contain preservative/fixatives such as in the CellSave [25] and Transfix [26] blood collection tubes frequently used for size-based enrichment procedures. These preservatives/fixatives make the cells more rigid facilitating the capture in the micropores. A potential solution would be to decrease the pore size of 5µm to enable the capture of the smaller CTC. The reduction of leukocytes will however need to be sufficient to not block the pores. The observed loss of spiked cultured tumor cells through the leukocyte depletion protocol of 26 and 29% suggest that a similar portion of CTC might be loss. Alternatively, a positive enrichment procedure such as the ones used in the CellSearch Profile kit [27], MagSweeper [28], EasySep™ [29], Dynal beads [29] and EpCAM independent methods [30], can be used to enrich CTC and pass them through the microwells. In one patient we obtained a second blood sample collected in an EDTA vacutainer and processed it similarly except we changed the staining and added fixation of the cells before placing the enriched sample on the microwells. The CTC recovery increase from 2 to 27% showing that the increased rigidity and higher fluorescent signals obtained from the Cytokeratin versus the EpCAM staining.

Viability of the CTC ranged from 0–33% (mean 13%), which is not surprising as many CTC detected with the CellSearch system show apoptotic features [31,32,33]. Some of the viable CTC remained viable for a couple of days but none divided. One of our aims is to measure the secretome of CTC and one the most important precondition for this to work is the availability of CTC that remain viable for a couple of days. Previously we have already shown that we can measure products secreted by individual cells present in the self-sorting microwells [34].

Other systems are available to assess single cells, for example a self-sufficient micro-droplet generation system that facilitates encapsulation, chemical stimulation, and microscopic analysis of viable cells inside droplets [35]. However, many cells are used and it has not been shown that such systems work for rare cell populations like CTC single cell isolation technologies suitable for CTC analysis, including viable cells do exist like DEPArray™ [36], ALS cellselector™ [37] or FACS sorting [38] but no successful CTC cultures have been published yet. A workflow to successfully obtain viable CTC using self-sorting microwells is presented here. The present configuration of the microwells, the pre-enrichment of CTC before placing the suspension on the sieves and the fluorescent labeling must be improved for a more efficient use of this workflow.

## 4. Materials and Methods 

### 4.1. Cell Culture

The human breast cancer cell lines MCF-7 (ATCC^®^ HTB-22^™^) and MDA-MB-231 (ATCC^®^ HTB-26^™^) were cultured in polystyrene flasks using Dulbecco’s modified Eagle medium (DMEM, Sigma, St. Louis, MO, USA) with 10% fetal bovine serum (FBS, Greiner Bio-One, Essen, Germany), 1000 IU/L penicillin and 1 mg/L streptomycin and 4mM L-Glutamine (Gibco, Thermo Fischer Scientific, Waltham, MA, USA). When flasks showed 80–90% confluence cells were trypsinized (0.05% trypsin-EDTA, Gibco) and replated in fresh culture flasks and incubated at 37 °C at 5% CO_2_. For experiments flasks with 70% confluence were used.

### 4.2. Viability Assay

MCF-7 cells were labeled with Calcein AM (Calc AM, live cells) and Ethidium homodimer 1 (EthD1, dead cells) (Live/Dead assay life Technologies, Invitrogen, Carlsbad, CA, USA) prior to seeding in microwells. Cells were harvested, washed once with 1× phosphate buffered saline (PBS) and finally diluted in 1 mL PBS. Subsequently cells were fluorescently labeled with Calc AM and EthD1 (1:10 *v*/*v*). Cells were incubated with live/dead staining solution in 100 µL for 20 min at 37 °C. Finally, 6000 cells were seeded in microwells and imaged using an automated fluorescence microscope (VyCAP, Deventer, The Netherlands) and cells were incubated in petri-dishes at 37 °C and 5% CO_2_. To determine cell viability in microwells overtime (4 h, 1 day, 2 days and 4 days), the cells were stained in the microwells. Here a sponge (VyCAP) was used, which enabled the removal of the cell culture medium from the bottom of the microwells. Once all medium was removed, 50 µL of Live/Dead mixture was added on top of the microwells and cells were incubated for 30 min at 37 °C. Microwells were scanned directly to determine the number of live and dead cells. Cells seeded by fluorescent activated cell sorting (FACS) and a manual pipette using a dilution series were used as a control.

### 4.3. Microwell Degassing and Sterilization

Prior to cell seeding, microwells (VyCAP) were degassed in 1× PBS with 0.1% Tween at a pressure of −0.5 bars for 15 min. Subsequently, the microwells were sterilized in 70% ethanol for 30 min and washed with 1× PBS to remove ethanol and finally incubated with cell culture medium for 30 min at 37 °C in the incubator. 

### 4.4. Cell Punching

For live cell punching of viable cells for further growth in 96 well tissue culture plates, microwells were scanned. From the acquired fluorescence images the wells that contain viable cells were selected by using the punching software program (VyCAP). Cells were retrieved from the microwells by punching the SiNi well bottoms from the microwells. Single cells or colonies were stained for viability and scanned. From the obtained fluorescent images viable cells were selected. Before punching the cells, fluid contact between the 96 well plate and the bottom of the microwells chip was made using cell culture medium. After establishing liquid contact and selection of the cells needed to be punched, the punch-software directs the needle to the selected microwells and punches the membrane containing the cell into the indicated well of the 96 well plate. The cell diffuses by gravity and hydrodynamic forces towards the bottom of the well and settles down. In some experiments, cells were left to grow for two days in the microwells and only those that underwent cell division were punched. In other experiments cells were punched after seeding into culture well plates. After punching, the number of cells was counted and followed in time to determine the number of cells undergoing cell division.

### 4.5. FACS Sorting

For fluorescent activated cell sorting (FACS) of cells a FACS ARIA II (BD Biosciences, San Jose, CA, USA) was used. Cells were stained using the Live/Dead assay as described above. Cells were sorted into 96 well culture plates (CellStar, Greiner). The instrument was calibrated using CS&T beads (BD Biosciences) before use. Before sorting, the sort gates and the number of cells to be sorted into the wells were set.

### 4.6. Manual Pipetting

Cells were harvested and counted using a Luna cell counter (Logos Biosystems, Westburg, Leusden, The Netherlands). Cells were diluted to a concentration of 1 cell/µL. A volume of 1 µL of this cell suspension was pipetted into a 96 wells culture plate. Using a fluorescence microscope the presence of the cell in a well was confirmed.

### 4.7. Spiking and Enrichment 

MCF-7 and MDA-MB-231 cells were pre-stained with CellTrace Violet (1:5000) (Invitrogen) and a number of 300 cells was spiked in 7.5 mL of EDTA anticoagulated whole blood from anonymized healthy volunteers obtained through the TNW-ECTM-donor services at the University of Twente. The research does not fall within the scope of the Dutch Medical Research Involving Human Subjects Act. Informed consent was obtained from all volunteers and blood collection procedures were approved by the local Medical Research Ethics Committee. Leukocytes were depleted from the blood by the RosetteSep™ Human CD45 Depletion kit (Stemcell Technologies, Cologne, Germany) that was used according to manufacturer’s instructions. The enriched fraction of tumor cells was gently collected and washed in PBS. Subsequently, this fraction was incubated with Alexa647 conjugated α-EpCAM (1:50 *v*/*v*, and PerCP conjugated α-CD45+ and with mixture of viability dye (Calc AM and EthD1) for 30 min at room temperature, the volume was increased by adding 1 mL of culture medium. Finally, the sample was seeded in microwells and imaged. Tumor cells were considered positive for respectively EpCAM, CellTrace Violet, and negative for CD45. Viable tumor cells were Calc AM positive, EthD1 negative and dead tumor cells Calc AM negative, EthD1 positive. Recoveries were determined as the number of counted tumor cells/number of spiked tumor cells * 100%. Optimal tumor cells were selected and punched out from microwells into a 96 wells plate. Microwell chip and the 96 wells plate were then incubated at 37 °C and 5% CO_2_. 

### 4.8. Patient Samples

Blood was drawn in CellSave vacutainers (Menarini Silicon Biosystems, Huntingdon Valley, PA, USA) for CellSearch^®^ analysis and screened for the presence of 10 or more CTC in 7.5 mL of blood at the Erasmus Medical Centre in Rotterdam, The Netherlands. From two metastatic breasts and four castration resistant prostate cancer patients with 10 or more CTC by CellSearch analysis blood, an additional blood tube was collected in EDTA vacutainers for self-sorting microwell analysis. Patients provided written informed consent and protocols approved by the Erasmus Ethics Committee (ethics reference number: MEC-17-238). 7.5 mL of blood samples for self-sorting microwell analysis were processed the next day. 7.5 mL of the sample was processed using the RosetteSep™ Human CD45 depletion kit according to the manufacturer’s instructions. The collected fraction was washed with 1× PBS and subsequently stained using α-EpCAM-Alexa647, α-CD45-PerCP, Calc AM and EthD1 and Hoechst 33342 for 30 min at 37 °C. Next 1 mL of cell culture medium was added and the sample was filtered onto a sterilized and degassed microwell chip by adding the sample into the chip and applying a pressure ranging from −10 to −70 mbar pressure. The microwells were scanned using a fluorescence microscope as soon as possible. After scanning, the microwell chip was kept in culture for three days, by placing the chip in culture medium and incubation at 37 °C at 5% CO_2_. The chip was checked on a daily basis for any contamination and scanned to monitor cell viability.

## Figures and Tables

**Figure 1 ijms-20-00477-f001:**
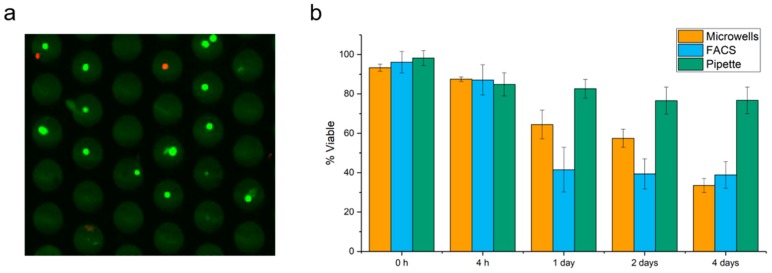
(**a**) Microscopic image of MCF-7 cells seeded into microwells. Live cells, Calc AM positive (green) Ethidium homodimer-1 (EthD1) negative or dead cells EthD1 positive (red), Calc AM negative. **(b**) Viability of single cells obtained by self-sorting microwells, fluorescent activated cells sorting (FACS) and pipette.

**Figure 2 ijms-20-00477-f002:**
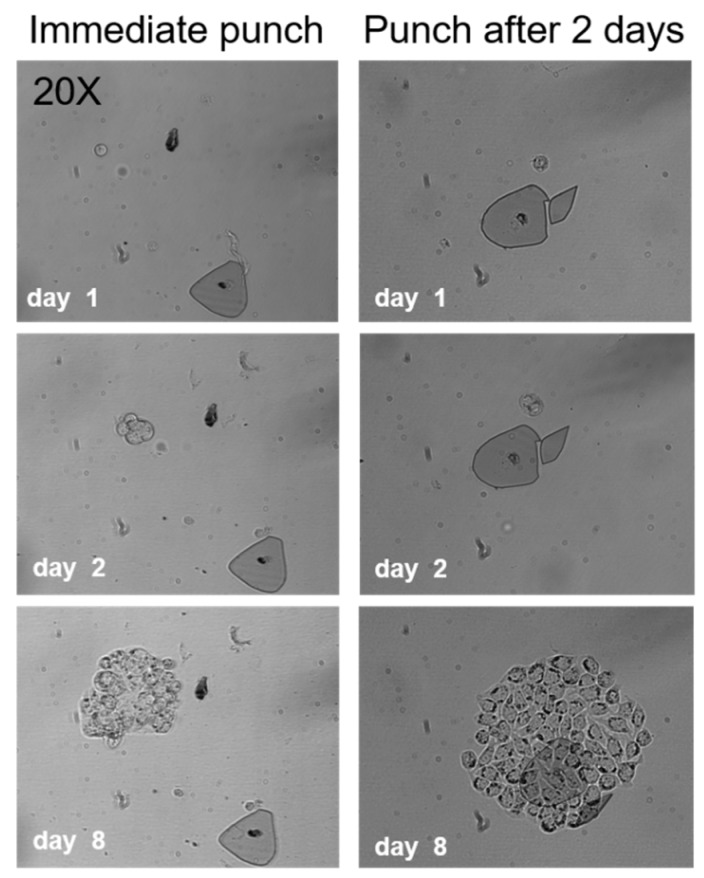
Growth pattern of single cells expanding after being punched from microwells into wells of a 96 well culture plate immediately after seeding (left) or two days after seeding (right), 20× magnification was used for imaging.

**Figure 3 ijms-20-00477-f003:**
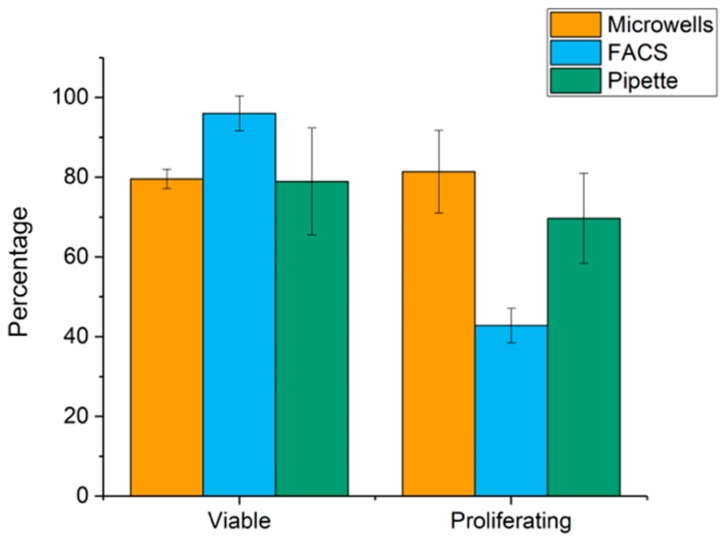
Cell viability and colony formation after punching compared to FACS sorting and pipetting. Percentage of viable cells and the percentage of cells that formed colonies is shown.

**Figure 4 ijms-20-00477-f004:**
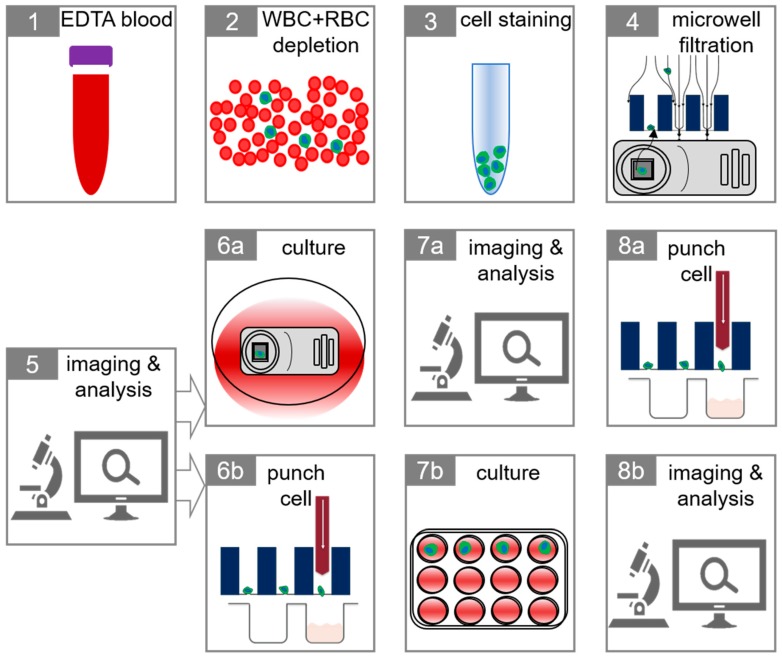
Schematic figure describing every step of our workflow to isolate and identify circulating tumor cells (CTC) from whole blood and subsequent cell culture. 7.5 mL of EDTA blood (1) is depleted of white blood cells (WBC) and erythrocytes (2). The enriched cells are then stained (3) using a live/dead assay, nuclear stain, WBC marker (CD45) and a CTC marker (EpCAM). Cells are filtered into the microwells (4) and imaged using a microscope and analyzed to identify viable CTC and dead CTC (5). Next, the microwell is either placed in culture (6a) or cells of interest are isolated (6b). After culture of the microwell plate (6a), they can be reexamined (7a) and the cells of interest isolated (8a). Following direct isolation (6b) the cells can be placed in the desired place for cell culture (7b) and imaged and analyzed at any time (8b).

**Figure 5 ijms-20-00477-f005:**
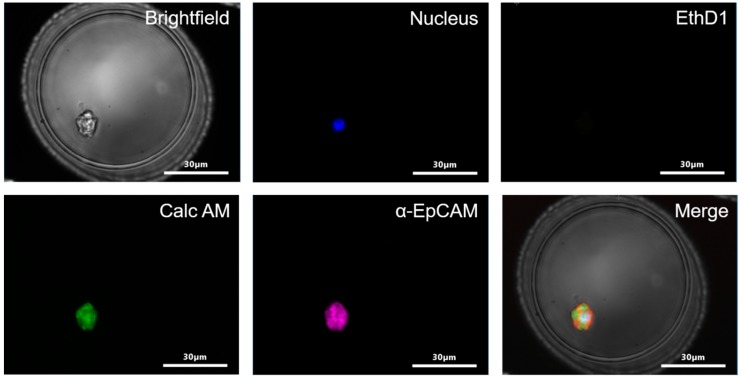
Identification of MCF-7 cell in a microwell. Bright field, Hoechst 33342 (nucleus) (blue), EthD1 to identify death cells (red), Calcein AM (Calc AM) to identify viable cells (green), EpCAM-Alexa647 to identify cells of epithelial origin (purple). Not shown is the CD45-PerCP image to identify leukocytes.

**Figure 6 ijms-20-00477-f006:**
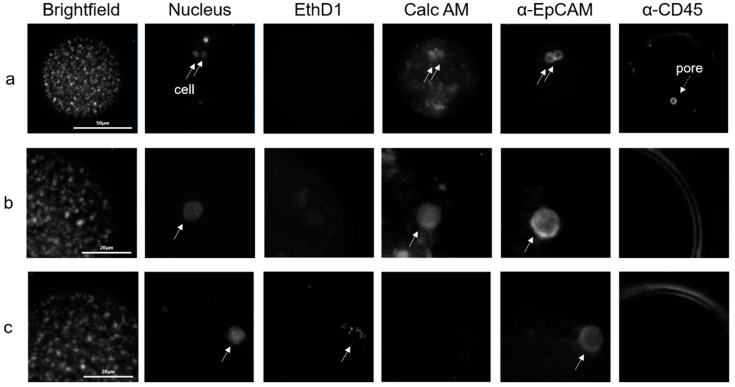
Images of CTCs in three microwells (Panels a, b & c) after isolation from 7.5 mL blood of a metastatic prostate cancer patient. The first image in the panels shows a bright field image showing the contours of the microwell, the objects in the image are residual red blood cells that end up in the well after blockage of the pore. Panel A shows the images of a CTC cluster consisting of two viable CTC (Nucleus+, EthD1-, Cals AM+, α-EpCAM+, α-CD45-). Panel B shows an example of a single viable CTC (Nucleus+, EthD1-, Cals AM+, α-EpCAM+, α-CD45-) and panel C shows a dead CTC (Nucleus+, EthD1+, Cals AM-, α-EpCAM+, α-CD45-). The solid arrows point to the position of the cells and the dashed arrow in the right image of panel A shows the position of the pore in the microwell visible through the PerCP filter cube.

**Table 1 ijms-20-00477-t001:** Recovery and viability after processing 7.5 mL EDTA blood spiked with ~100 MCF-7 and ~100 MDA-MB-231 cells. After 2 days of cell culture the percentage of cells that showed cell growth was determined. In two of the experiments an infection was observed (inf).

	% Recovery		% Viable		% Grow	
	MCF-7	MDA-MB-231	MCF-7	MDA-MB-231	MCF-7	MDA-MB-231
1	61	67	63	78	55	49
2	47	51	61	67	55	48
3	43	39	74	65	64	inf
4	45	49	75	65	61	44
5	33	38	72	77	59	inf
6	30	41	58	65	69	54
mean	43	48	67	70	59	49
SD	11	11	7	6	4	4

**Table 2 ijms-20-00477-t002:** Number of CTC in 5 metastatic breast and 2 prostate cancer patients detected in 7.5 mL of blood by CellSearch^®^ and in the microwells after CD45 depletion of 7.5 mL of blood and passage into the self-sorting microwells. Shown is the recovery in the microwells compared to CellSearch, also the percentage of cells that are viable inside the mcirowells. In addition the percentage of viable cells still showing viability after two days. NA is not applicable.

	Cancer	CellSearch^®^ CTC	Microwell CTC	Recovery%	% Viable	% Viable after 2 Days
1	Breast	102	1	1	0	0
2	Breast	>100	0	0	0	0
3	Prostate	498	22	4	36	25
4	Breast	94	3	3	33	0
5	Breast	22	0	0	0	0
6	Breast	111	5	5	20	100
7a	Prostate	55	1	2	0	0
mean		140	5	2	13	18
SD		161	8	2	17	37
7b *	Prostate	55	15	27	NA	NA

* An additional blood sample obtained from patient 7 was processed identically except for the staining, which now included Cytokeratin for which cell permeabilization and fixation was needed before passage onto the self-sorting microwells.

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
