# Peer review of "Self-Seeding Microwells to Isolate and Assess the Viability of Single Circulating Tumor Cells"

_ijms, 2019, doi:10.3390/ijms20030477_

Round 1
Reviewer 1 Report
The manuscript entitled "Self-Seeding Microwells to Isolate and Assess the Viability of Single Circulating Tumor Cells" is very interesting. I would suggest the manuscript to be accepted following a minor revision, addressing my following comments:
Please expand the introduction, and further explain the importance of your work.
Please add a schematic figure, describing every step of your experimental process, including 'punching' and the experimental setup such that the reader can understand the entire work through this figure.
Please explain the limitations of your method against the diameter of cells or their tendency to adhere to the surface.
Please discuss and properly cite the following work, which describes a self-sufficient droplet generation method for storage and microscopic analysis of cells:
A self-sufficient micro-droplet generation system using highly porous elastomeric sponges: A versatile tool for conducting cellular assays, Sensors and Actuators B: Chemical, 2018, 274, 645-653
Author Response
We would like to thank the reviewer for reviewing our manuscript and the given comments and suggestions. Please find our reply to the reviewers comments below.
The manuscript entitled "Self-Seeding Microwells to Isolate and Assess the Viability of Single Circulating Tumor Cells" is very interesting. I would suggest the manuscript to be accepted following a minor revision, addressing my following comments:
Please expand the introduction, and further explain the importance of your work.
- We improved and extended our introduction by reviewing additional papers describing organoid culture from tumor biopsies and currently existing CTC culture papers. We discussed the possibilities and feasibility of starting CTC culture models. Also we discussed and explained more the importance of our work.
The following paraphs were added: ‘ To date protocols [10] are available for the culture of organoids from tumor biopsies to be used as a model for disease [11,12]. However, culturing CTC has only been demonstrated by a few groups. Kolostova et al. used a size based separation method to isolate and culture urothelial CTC directly on the separation membrane [13]. Second, Cayrefourcq et al. negatively enriched blood samples from 71 patients with metastatic colon cancer patients and successfully established one permanent cell line from a patient having a CTC count of ≥300 [14]. In prostate cancer, Gao et al. succeeded in establishing a 3D organoid system for the long-term culture of CTC derived from peripheral blood of castration resistant metastatic patients [15]. The group of Nagrath focused on lung cancer and developed a novel in situ capture and culture methodology for ex vivo expansion of CTC using a 3D co-culture model. CTC were successfully expended from 14 of 19 early stage lung cancer patients, using a three dimensional co-culture model, including fibroblasts, to support tumor development [16].
In addition, very recently several groups looked into the use of leukapheresis products, as a source for CTC, where the likelihood of finding of CTC is greater and were able to establish CTC cultures in mouse models [17,18]. These discussed examples indicate that the establishment of functional CTC cell line models is feasible. The isolation and in vitro culture of CTC may provide an opportunity to noninvasively monitor the changing patterns of drug resistance in individual patients while their tumors acquire new mutations and might improve treatment.’
‘Also, when interested in CTC heterogeneity what is lacking in the studies discussed above is the possibility to assess molecular heterogeneity between CTC within an individual patient. Cultures are established from the bulk of CTC isolated from patients making it harder to assess their heterogeneity. Single cell isolation techniques might contribute to this demand.‘
Please add a schematic figure, describing every step of your experimental process, including 'punching' and the experimental setup such that the reader can understand the entire work through this figure.
- We added a schematic figure describing every step of our workflow to isolate and identify individual CTC from whole blood with subsequent culture nd viability assessment.
The following was added:
Figure 4: Schematic figure describing every step of our workflow to isolate and identify CTC from whole blood and subsequent cell culture. 7.5 mL of EDTA blood (1) is depleted of WBC and erythrocytes (2). The enriched cells are then stained (3) using a live/dead assay, nuclear stain, WBC marker (CD45) and a CTC marker (EpCAM). Cells are filtered into the microwells (4) and imaged using a microscope and analyzed to identify viable CTC and dead CTC (5). Next, the microwell is either placed in culture (6a) or cells of interest are isolated (6b). After culture of the microwell plate (6a), they can be reexamined (7a) and the cells of interest isolated (8a). Following direct isolation (6b) the cells can be placed in the desired place for cell culture (7b) and imaged and analyzed at any time (8b).
Please explain the limitations of your method against the diameter of cells or their tendency to adhere to the surface.
- We extended our discussion on cell diameter by adding this paragraph:
‘CTC are generally larger in size then WBC, and this difference is used in the microwell filtration step to remove remaining WBC and RBC after depletion and obtain individual CTC in the microwells. However, where the size of cells from tumor cell lines is known, the actual size of CTC in individual patient is unknown and might cover a wide range [26]. In addition to size, also cell stiffness or rigidity has an important role in passage through a pore [27], cell stiffness of patient CTC may vary and be less than in cell lines, this might explain part of the cell loss.’
(26. Stoecklein, N.H.; Fischer, J.C.; Niederacher, D.; Terstappen, L.W.M.M. Challenges for CTC-based liquid biopsies: low CTC frequency and diagnostic leukapheresis as a potential solution. Expert Rev. Mol. Diagn. 2016, 16, 147–164, doi:10.1586/14737159.2016.1123095.
27. Coumans, F.A.W.; van Dalum, G.; Beck, M.; Terstappen, L.W.M.M. Filtration Parameters Influencing Circulating Tumor Cell Enrichment from Whole Blood. PLoS One 2013, 8, e61774, doi:10.1371/journal.pone.0061774)
Please discuss and properly cite the following work, which describes a self-sufficient droplet generation method for storage and microscopic analysis of cells:
A self-sufficient micro-droplet generation system using highly porous elastomeric sponges: A versatile tool for conducting cellular assays, Sensors and Actuators B: Chemical, 2018, 274, 645-653
- We expanded our discussion by discussing more references which make use of single cell isolation techniques. We also added the reference as mentioned by the reviewer as one of the discussed references. The following paragraph was added:
‘Other systems are available to assess single cells, for example a self-sufficient micro-droplet generation system which facilitates encapsulation, chemical stimulation and microscopic analysis of viable cells inside droplets [38]. However many cells are used and it has not been shown that such systems work for rare cell populations like CTC, Single cell isolation technologies suitable for CTC analysis, including viable cells do exist like DEPArray™ [39], ALS cellselector™ [40] or FACS sorting [41] but no successful CTC cultures have been published yet.’
Reviewer 2 Report
This manuscript cannot be acceptable for publication without consideration of major criticism points.
The submitted manuscript reports the evaluation of isolation and assessment of the viability of circulating tumor cells (CTC) using a self-seeding microwell device. Although the manuscript is interesting report in the research field of CTC analysis, the interpretation is not enough for explanation about the difference and advantage of the system compared with conventional and reported microchip devices for CTC microchips because there are a lot of reports about CTC isolation and analyses microchip devices. And also, authors showed MCF-7 cells and CTCs cultivation in the microwells devices. However, further futuristic applications and advantage of the system is not clear compared with conventional and other reported systems. And also, why the microwells system showed higher viability of CTC compared with conventional systems? So, it is necessary to clearly describe difference and advantage of the microwells system compared with other reported research. Therefore, this manuscript cannot be acceptable for publication without consideration of major criticism points.
Author Response
We would like to thank the reviewer for reviewing our manuscript and the given comments and suggestions. Please find our reply to the reviewers comments below.
The submitted manuscript reports the evaluation of isolation and assessment of the viability of circulating tumor cells (CTC) using a self-seeding microwell device. Although the manuscript is interesting report in the research field of CTC analysis, the interpretation is not enough for explanation about the difference and advantage of the system compared with conventional and reported microchip devices for CTC microchips because there are a lot of reports about CTC isolation and analyses microchip devices.
- We extended our introduction by reviewing additional papers describing organoid culture from tumor biopsies and currently existing CTC culture papers to compare to currently existing technologies including microfluidic chips.
The following was added to our introduction: ‘To date protocols [10] are available for the culture of organoids from tumor biopsies to be used as a model for disease [11,12]. However, culturing CTC has only been demonstrated by a few groups. Kolostova et al. used a size based separation method to isolate and culture urothelial CTC directly on the separation membrane [13]. Second, Cayrefourcq et al. negatively enriched blood samples from 71 patients with metastatic colon cancer patients and successfully established one permanent cell line from a patient having a CTC count of ≥300 [14]. In prostate cancer, Gao et al. succeeded in establishing a 3D organoid system for the long-term culture of CTC derived from peripheral blood of castration resistant metastatic patients [15]. The group of Nagrath focused on lung cancer and developed a novel in situ capture and culture methodology for ex vivo expansion of CTC using a 3D co-culture model. CTC were successfully expended from 14 of 19 early stage lung cancer patients, using a three dimensional co-culture model, including fibroblasts, to support tumor development [16].
In addition, very recently several groups looked into the use of leukapheresis products, as a source for CTC, where the likelihood of finding of CTC is greater and were able to establish CTC cultures in mouse models [17,18]. These discussed examples indicate that the establishment of functional CTC cell line models is feasible. The isolation and in vitro culture of CTC may provide an opportunity to noninvasively monitor the changing patterns of drug resistance in individual patients while their tumors acquire new mutations and might improve treatment.’….
And also, authors showed MCF-7 cells and CTCs cultivation in the microwells devices. However, further futuristic applications and advantage of the system is not clear compared with conventional and other reported systems.
- We a tried to explain that our workflow has the advantage of assessing heterogeneity between CTC, which is lacking on other papers. We added the following paragraph in the introduction to discuss this:
‘… Also, when interested in CTC heterogeneity what is lacking in the studies discussed above is the possibility to assess molecular heterogeneity between CTC within an individual patient. Cultures are established from the bulk of CTC isolated from patients making it harder to assess their heterogeneity. Single cell isolation techniques might contribute to this demand. Here we use the previously introduced self-sorting microwells [19,20] to establish methods to discriminate between individual viable and non-viable cancer cells and establish conditions to maintain the viability of the cancer cells.’
-
And also, why the microwells system showed higher viability of CTC compared with conventional systems? So, it is necessary to clearly describe difference and advantage of the microwells system compared with other reported research. Therefore, this manuscript cannot be acceptable for publication without consideration of major criticism points.
- We extended our discussion to address the final point of the reviewer: ‘why the microwells system showed higher viability of CTC compared with conventional systems’
We don’t believe shows higher viability than other systems. We believe the advantage of our system is the fast and easy workflow and especially the possibility to obtain pure individual CTC allowing for the assessment of heterogeneity. The following was added: ‘Other systems are available to assess single cells, for example a self-sufficient micro-droplet generation system which facilitates encapsulation, chemical stimulation and microscopic analysis of viable cells inside droplets [38]. However many cells are used and it has not been shown that such systems work for rare cell populations like CTC, Single cell isolation technologies suitable for CTC analysis, including viable cells do exist like DEPArray™ [39], ALS cellselector™ [40] or FACS sorting [41] but no successful CTC cultures have been published yet.’
Round 2
Reviewer 2 Report
The submitted manuscript reports the evaluation of isolation and assessment of the viability of circulating tumor cells (CTC) using a self-seeding microwell device. The authors explained advantage of the micro microwell device system in the revised manuscript based on suggestion from the reviewer. Therefore the manuscript can be acceptable for publication.